# A Probabilistic U-Net for Segmentation of Ambiguous Images

**Simon A. A. Kohl**[1,*,2,], **Bernardino Romera-Paredes**[1], **Clemens Meyer**[1], **Jeffrey De Fauw**[1], **Joseph R. Ledsam**[1], **Klaus H. Maier-Hein**[2], **S. M. Ali Eslami**[1], **Danilo Jimenez Rezende**[1], and **Olaf Ronneberger**[1]

[1]DeepMind, London, UK
[2]Division of Medical Image Computing, German Cancer Research Center, Heidelberg, Germany
{simon.kohl,k.maier-hein}@dkfz.de
{brp,meyerc,defauw,jledsam,aeslami,danilor,olafr}@google.com

## Abstract

Many real-world vision problems suffer from inherent ambiguities. In clinical applications for example, it might not be clear from a CT scan alone which particular region is cancer tissue. Therefore a group of graders typically produces a set of diverse but plausible segmentations. We consider the task of learning a distribution over segmentations given an input. To this end we propose a generative segmentation model based on a combination of a U-Net with a conditional variational autoencoder that is capable of efficiently producing an unlimited number of plausible hypotheses. We show on a lung abnormalities segmentation task and on a Cityscapes segmentation task that our model reproduces the possible segmentation variants as well as the frequencies with which they occur, doing so significantly better than published approaches. These models could have a high impact in real-world applications, such as being used as clinical decision-making algorithms accounting for multiple plausible semantic segmentation hypotheses to provide possible diagnoses and recommend further actions to resolve the present ambiguities.

## 1 Introduction

The semantic segmentation task assigns a class label to each pixel in an image. While in many cases the context in the image provides sufficient information to resolve the ambiguities in this mapping, there exists an important class of images where even the full image context is not sufficient to resolve all ambiguities. Such ambiguities are common in medical imaging applications, e.g., in lung abnormalities segmentation from CT images. A lesion might be clearly visible, but the information about whether it is cancer tissue or not might not be available from this image alone. Similar ambiguities are also present in photos. E.g. a part of fur visible under the sofa might belong to a cat or a dog, but it is not possible from the image alone to resolve this ambiguity[2]. Most existing segmentation algorithms either provide only one likely consistent hypothesis (e.g., "all pixels belong to a cat") or a pixel-wise probability (e.g., "each pixel is 50% cat and 50% dog").

Especially in medical applications where a subsequent diagnosis or a treatment depends on the segmentation map, an algorithm that only provides the most likely hypothesis might lead to misdiagnoses

---

[*]work done during an internship at DeepMind.

[2]In [1] this is defined as *ambiguous evidence* in contrast to *implicit class confusion*, that stems from an ambiguous class definition (e.g. the concepts of desk vs. table). For the presented work this differentiation is not required.

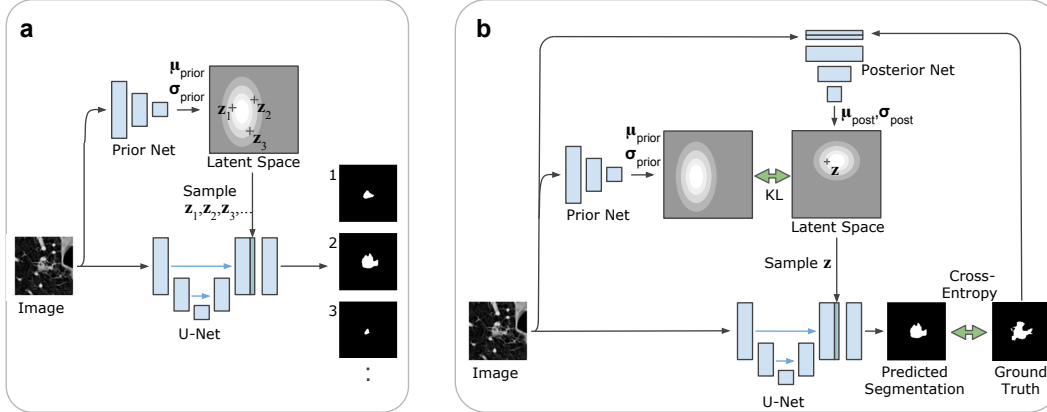

Figure 1: The Probabilistic U-Net. (**a**) Sampling process. Arrows: flow of operations; blue blocks: feature maps. The heatmap represents the probability distribution in the low-dimensional latent space $\mathbb{R}^N$ (e.g., $N = 6$ in our experiments). For each execution of the network, one sample $\mathbf{z} \in \mathbb{R}^N$ is drawn to predict one segmentation mask. Green block: $N$-channel feature map from broadcasting sample $\mathbf{z}$. The number of feature map blocks shown is reduced for clarity of presentation. (**b**) Training process illustrated for one training example. Green arrows: loss functions.

and sub-optimal treatment. Providing only pixel-wise probabilities ignores all co-variances between the pixels, which makes a subsequent analysis much more difficult if not impossible. If multiple consistent hypotheses are provided, these can be directly propagated into the next step in a diagnosis pipeline, they can be used to suggest further diagnostic tests to resolve the ambiguities, or an expert with access to additional information can select the appropriate one(s) for the subsequent steps.

Here we present a segmentation framework that provides multiple segmentation hypotheses for ambiguous images (Fig. 1a). Our framework combines a conditional variational auto encoder (CVAE) [2, 3, 4, 5] which can model complex distributions, with a U-Net [6] which delivers state-of-the-art segmentations in many medical application domains. A low-dimensional latent space encodes the possible segmentation variants. A random sample from this space is injected into the U-Net to produce the corresponding segmentation map. One key feature of this architecture is the ability to model the joint probability of all pixels in the segmentation map. This results in multiple segmentation maps, where each of them provides a consistent interpretation of the whole image. Furthermore our framework is able to also learn hypotheses that have a low probability and to predict them with the corresponding frequency. We demonstrate these features on a lung abnormalities segmentation task, where each lesion has been segmented independently by four experts, and on the Cityscapes dataset, where we artificially flip labels with a certain frequency during training.

A body of work with different approaches towards probabilistic and multi-modal segmentation exists. The most common approaches provide independent pixel-wise probabilities [7, 8]. These models induce a probability distribution by using dropout over spatial features. Whereas this strategy fulfills this line of work's objective of quantifying the pixel-wise uncertainty, it produces inconsistent outputs. A simple way to produce plausible hypotheses is to learn an ensemble of (deep) models [9]. While the outputs produced by ensembles are consistent, they are not necessarily diverse and ensembles are typically not able to learn the rare variants as their members are trained independently. In order to overcome this, several approaches train models jointly using the oracle set loss [10], i.e. a loss that only accounts for the closest prediction to the ground truth. This has been explored in [11] and [1] using an ensemble of deep networks, and in [12] and [13] using one common deep network with $M$ heads. While multi-head approaches may have the capacity to capture a diverse set of variants, they are not equipped to learn the occurrence frequencies of individual variants. Two common disadvantages of both ensembles and $M$ heads models are their ungraceful scaling to large numbers of hypotheses, and their requirement of fixing the number of allowed hypotheses at training time. Another set of approaches to produce multiple diverse solutions relies on graphical models, such as junction chains [14], and more generally Markov Random Fields [15, 16, 17, 18]. While many of the

previous approaches are guaranteed to find the best diverse solutions, these are confined to structured problems whose dependencies can be described by tractable graphical models.

The task of image-to-image translation [19] tackles a very similar problem: an under-constrained domain transfer of images needs to be learned. Many of the recent approaches employ generative adversarial networks (GANs) which are known to suffer from challenges such as 'mode-collapse' [20]. In an attempt to solve the mode-collapse problem, the 'bicycleGAN' [21] involves a component that is similar in architecture to ours. In contrast to our proposed architecture, their model encompasses a fixed prior distribution and during training their posterior distribution is only conditioned on the output image. Very recent work on generating appearances given a shape encoding [22] also combines a U-Net with a VAE, and was developed concurrently to ours. In contrast to our proposal, their training requires an additional pretrained VGG-net that is employed as a reconstruction loss. Finally, in [23] is proposed a probabilistic model for structured outputs based on optimizing the dissimilarity coefficient [24] between the ground truth and predicted distributions. The resultant approach is assessed on the task of hand pose estimation, that is, predicting the location of 14 joints, arguably a simpler space compared to the space of segmentations we consider here. Similarly to the approach presented below, they inject latent variables at a later stage of the network architecture.

The main contributions of this work are: (1) Our framework provides consistent segmentation maps instead of pixel-wise probabilities and can therefore give a joint likelihood of modes. (2) Our model can induce arbitrarily complex output distributions including the occurrence of very rare modes, and is able to learn calibrated probabilities of segmentation modes. (3) Sampling from our model is computationally cheap. (4) In contrast to many existing applications of deep generative models that can only be qualitatively evaluated, our application and datasets allow quantitative performance evaluation including penalization of missing modes.

## 2 Network Architecture and Training Procedure

Our proposed network architecture is a combination of a conditional variational auto encoder [2, 3, 4, 5] with a U-Net [6], with the objective of learning a conditional density model over segmentations, conditioned on the image.

**Sampling.** The central component of our architecture (Fig. 1a) is a low-dimensional latent space $\mathbb{R}^N$ (e.g., $N = 6$, which performed best in our experiments). Each position in this space encodes a segmentation variant. The 'prior net', parametrized by weights $\omega$, estimates the probability of these variants for a given input image $X$. This prior probability distribution (called $P$ in the following) is modelled as an axis-aligned Gaussian with mean $\boldsymbol{\mu}_{\text{prior}}(X; \omega) \in \mathbb{R}^N$ and variance $\boldsymbol{\sigma}_{\text{prior}}(X; \omega) \in \mathbb{R}^N$. To predict a set of $m$ segmentations we apply the network $m$ times to the same input image (only a small part of the network needs to be re-evaluated in each iteration, see below). In each iteration $i \in \{1, \ldots, m\}$, we draw a random sample $\mathbf{z}_i \in \mathbb{R}^N$ from $P$

$$\mathbf{z}_i \sim P(\cdot | X) = \mathcal{N}\left(\boldsymbol{\mu}_{\text{prior}}(X; \omega), \text{diag}(\boldsymbol{\sigma}_{\text{prior}}(X; \omega))\right) , \qquad (1)$$

broadcast the sample to an $N$-channel feature map with the same shape as the segmentation map, and concatenate this feature map to the last activation map of a U-Net (the U-Net is parameterized by weights $\theta$). A function $f_{\text{comb.}}$ composed of three subsequent $1 \times 1$ convolutions ($\psi$ being the set of their weights) combines the information and maps it to the desired number of classes. The output, $S_i$, is the segmentation map corresponding to point $\mathbf{z}_i$ in the latent space:

$$S_i = f_{\text{comb.}}\left(f_{\text{U-Net}}(X; \theta), \mathbf{z}_i; \psi\right) . \qquad (2)$$

Notice that when drawing $m$ samples for the same input image, we can reuse the output of the prior net and the feature activations of the U-Net. Only the function $f_{\text{comb.}}$ needs to be re-evaluated $m$ times.

**Training.** The networks are trained with the standard training procedure for conditional VAEs (Fig. 1b), i.e. by minimizing the variational lower bound (Eq. 4). The main difference with respect to training a deterministic segmentation model, is that the training process additionally needs to find a useful embedding of the segmentation variants in the latent space. This is solved by introducing a 'posterior net', parametrized by weights $\nu$, that learns to recognize a segmentation variant (given the raw image $X$ and the ground truth segmentation $Y$) and to map this to a position $\boldsymbol{\mu}_{\text{post}}(X, Y; \nu) \in \mathbb{R}^N$

with some uncertainty $\boldsymbol{\sigma}_{\text{post}}(X, Y; \nu) \in \mathbb{R}^N$ in the latent space. The output is denoted as posterior distribution $Q$. A sample $\mathbf{z}$ from this distribution,

$$\mathbf{z} \sim Q(\cdot|X, Y) = \mathcal{N}\left(\boldsymbol{\mu}_{\text{post}}(X, Y; \nu), \text{diag}(\boldsymbol{\sigma}_{\text{post}}(X, Y; \nu))\right), \tag{3}$$

combined with the activation map of the U-Net (Eq. 1) must result in a predicted segmentation $S$ identical to the ground truth segmentation $Y$ provided in the training example. A cross-entropy loss penalizes differences between $S$ and $Y$ (the cross-entropy loss arises from treating the output $S$ as the parameterization of a pixel-wise categorical distribution $P_c$). Additionally there is a Kullback-Leibler divergence $D_{\text{KL}}(Q||P) = \mathbb{E}_{z \sim Q}\left[\log Q - \log P\right]$ which penalizes differences between the posterior distribution $Q$ and the prior distribution $P$. Both losses are combined as a weighted sum with a weighting factor $\beta$, as done in [25]:

$$\mathcal{L}(Y, X) = \mathbb{E}_{z \sim Q(\cdot|Y, X)}\left[-\log P_c(Y|S(X, z))\right] + \beta \cdot D_{\text{KL}}\left(Q(z|Y, X)||P(z|X)\right). \tag{4}$$

The training is done from scratch with randomly initialized weights. During training, this KL loss "pulls" the posterior distribution (which encodes a segmentation variant) and the prior distribution towards each other. On average (over multiple training examples) the prior distribution will be modified in a way such that it "covers" the space of all presented segmentation variants for a specific input image[3].

## 3 Performance Measures and Baseline Methods

In this section we first present the metric used to assess the performance of all approaches, and then describe each competitor approach used in the comparisons.

### 3.1 Performance measures

As it is common in the semantic segmentation literature, we employ the intersection over union (IoU) as a measure to compare a pair of segmentations. However, in the present case, we not only want to compare a deterministic prediction with a unique ground truth, but rather we are interested in comparing distributions of segmentations. To do so, we use the *generalized energy distance* [26, 27, 28], which leverages distances between observations:

$$D_{\text{GED}}^2(P_{\text{gt}}, P_{\text{out}}) = 2\mathbb{E}\left[d(S, Y)\right] - \mathbb{E}\left[d(S, S')\right] - \mathbb{E}\left[d(Y, Y')\right], \tag{5}$$

where $d$ is a distance measure, $Y$ and $Y'$ are independent samples from the ground truth distribution $P_{\text{gt}}$, and similarly, $S$ and $S'$ are independent samples from the predicted distribution $P_{\text{out}}$. The energy distance $D_{\text{GED}}$ is a metric as long as $d$ is also a metric [29]. In our case we choose $d(x, y) = 1 - \text{IoU}(x, y)$, which as proved in [30, 31], is a metric. In practice, we only have access to samples from the distributions that models induce, so we rely on statistics of Eq. 5, $\hat{D}_{\text{GED}}^2$. The details about its computation for each experiment are presented in Appendix B.

### 3.2 Baseline methods

With the aim of providing context for the performance of our proposed approach we compare against a range of baselines. To the best of our knowledge there exists no other work that has considered capturing a distribution over multi-modal segmentations and has measured the agreement with such a distribution. For fair comparison, we train the baseline models whose architectures are depicted in Fig. 2 in the exact same manner as we train ours. The baseline methods all involve the same U-Net architecture, i.e. they share the same core component and thus employ comparable numbers of learnable parameters in the segmentation tasks.

**Dropout U-Net** (Fig. 2a). Our 'Dropout U-Net' baselines follow the Bayesian segnet's [7] proposition: we dropout the activations of the respective incoming layers of the three inner-most encoder and decoder blocks with a dropout probability of $p = 0.5$ during training as well as when sampling.

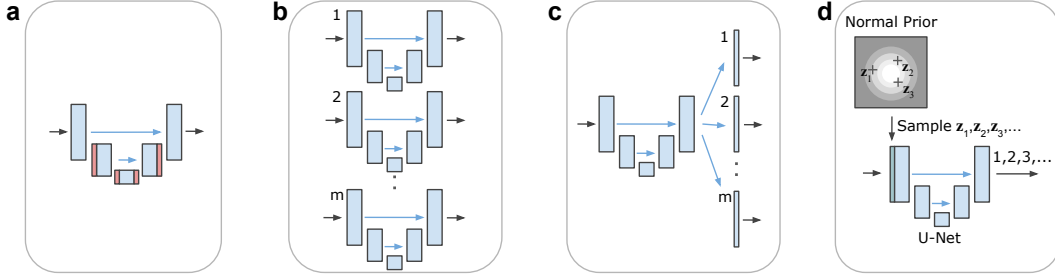

Figure 2: Baseline architectures. Arrows: flow of operations; blue blocks: feature maps; red blocks: feature maps with dropout; green block broadcasted latents. Note that the number of feature map blocks shown is reduced for clarity of presentation. (**a**) Dropout U-Net. (**b**) U-Net Ensemble. (**c**) M-Heads. (**d**) Image2Image VAE.

**U-Net Ensemble** (Fig. 2b). We report results for ensembles with the number of members matching the required number of samples (referred to as 'U-Net Ensemble'). The original deterministic variant of the U-Net is the 1-sample corner case of an ensemble.

**M-Heads** (Fig. 2c). Aiming for diverse semantic segmentation outputs, the works of [12] and [13] propose to branch off $M$ heads after the last layer of a deep net each of which contributes one output variant. An adjusted cross-entropy loss that adaptively assigns heads to ground-truth hypotheses is employed to promote diversity while reducing the risk of idle heads: the loss of the best performing head is weighted with a factor of $1 - \epsilon$, while the remaining heads each contribute with a weight of $\epsilon/(M - 1)$ to the loss. For our 'M-Heads' baselines we again employ a U-Net core and set $\epsilon = 0.05$ as proposed by [12]. In order to allow for the evaluation of 4, 8 and 16 samples, we train M-Heads models with the corresponding number of heads.

**Image2Image VAE** (Fig. 2d). In [21] the authors propose a U-Net VAE-GAN hybrid for multi-modal image-to-image translation, that owes its stochasticity to normal distributed latents that are broadcasted and fed into the encoder path of the U-Net. In order to deal with the complex solution space in image-to-image translation tasks, they employ an adversarial discriminator as additional supervision alongside a reconstruction loss. In the fully supervised setting of semantic segmentation such an additional learning signal is however not necessary and we therefore train with a cross-entropy loss only. In contrast to our proposition, this baseline, which we refer to as the 'Image2Image VAE', employs a prior that is not conditioned on the input image (a fixed normal distribution) and a posterior net that is not conditioned on the input either.

In all cases we examine the models' performance when drawing a different number of samples (1, 4, 8 and 16) from each of them.

# 4   Results

A quantitative evaluation of multiple segmentation predictions per image requires annotations from multiple labelers. Here we consider two datasets: The LIDC-IDRI dataset [32, 33, 34] which contains 4 annotations per input, and the Cityscapes dataset [35], which we artificially modify by adding *synonymous classes* to introduce uncertainty in the way concepts are labelled.

## 4.1   Lung abnormalities segmentation

The LIDC-IDRI dataset [32, 33, 34] contains 1018 lung CT scans from 1010 lung patients with manual lesion segmentations from four experts. This dataset is a good representation of the typical ambiguities that appear in CT scans. For each scan, 4 radiologists (from a total of 12) provided annotation masks for lesions that they independently detected and considered to be abnormal. We use the masks resulting from a second reading in which the radiologists were shown the anonymized annotations of the others and were allowed to make adjustments to their own masks.

For our experiments we split this dataset into a training set composed of 722 patients, a validation set composed of 144 patients, and a test set composed of the remaining 144 patients. We then resampled

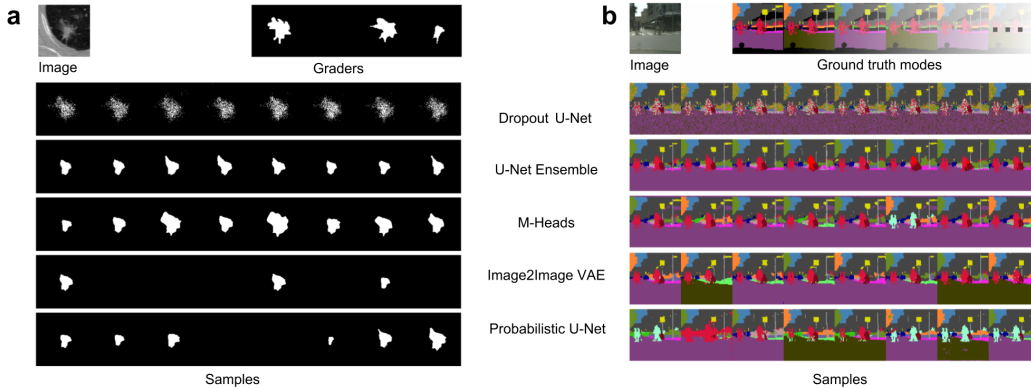

Figure 3: Qualitative results. The first row shows the input image and the ground truth segmentations. The following rows show results from the baselines and from our proposed method. (**a**) lung CT scan from the LIDC test set. Ground truth: 4 graders. (**b**) Cityscapes. Images cropped to squares for ease of presentation. Ground truth: 32 artificial modes. Best viewed in colour.

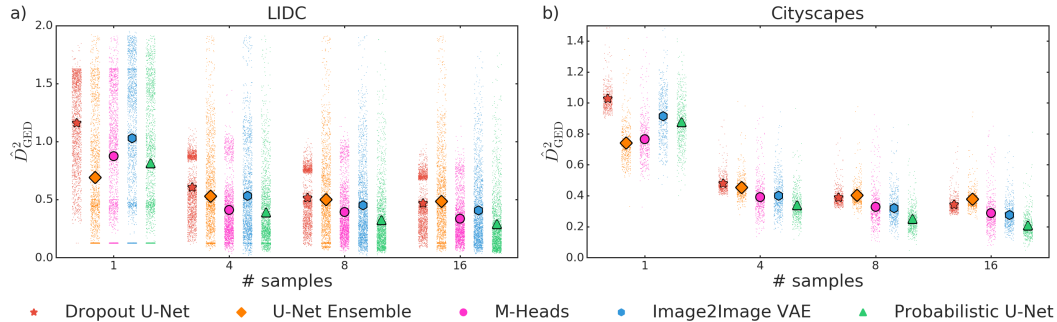

Figure 4: Comparison of approaches using the squared energy distance. Lower energy distances correspond to better agreement between predicted distributions and ground truth distribution of segmentations. The symbols that overlay the distributions of data points mark the mean performance. (**a**) Performance on lung abnormalities segmentation on our LIDC-IDRI test-set. (**b**) Performance on the official Cityscapes validation set (our test set).

the CT scans to $0.5\,\mathrm{mm} \times 0.5\,\mathrm{mm}$ in-plane resolution (the original resolution is between $0.461\,\mathrm{mm}$ and $0.977\,\mathrm{mm}$, $0.688\,\mathrm{mm}$ on average) and cropped 2D images ($180 \times 180$ pixels) centered at the lesion positions. The lesion positions are those where at least one of the experts segmented a lesion. By cropping the scans, the resultant task is in isolation not directly clinically relevant. However, this allows us to ignore the vast areas in which all labelers agree, in order to focus on those where there is uncertainty. This resulted in 8882 images in the training set, 1996 images in the validation set and 1992 images in the test set. Because the experts can disagree whether the lesion is abnormal tissue, up to 3 masks per image can be empty. Fig. 3a shows an example of such lesion-centered images and the masks provided by 4 graders.

As all models share the same U-Net core component and for fairness and ease of comparability, we let all models undergo the same training schedule, which is detailed in subsection H.1.

In order to grasp some intuition about the kind of samples produced by each model, we show in Fig. 3a, as well as in Appendix F, representative results for the baseline methods and our proposed Probabilistic U-Net. Fig. 4a shows the squared generalized energy distance $\hat{D}_{\mathrm{GED}}^2$ for all models as a function of the number of samples. The data accumulations visible as horizontal stripes are owed to the existence of empty ground-truth masks. The energy distance on the 1992 images large lung abnormalities test set, decreases for all models as more samples are drawn indicating an improved matching of the ground-truth distribution as well as enhanced sample diversity. Our proposed

Probabilistic U-Net outperforms all baselines when sampling 4, 8 and 16 times. The performance at 16 samples is found significantly higher than that of the baselines ($p$-value $\sim \mathcal{O}(10^{-13})$), according to the Wilcoxon signed-rank test. Finally, in Appendix E we show the results of an experiment regarding the capacity different models have to distinguish between unambiguous and ambiguous instances (i.e. instances where graders disagree on the presence of a lesion).

## 4.2 Cityscapes semantic segmentation

As a second dataset we use the Cityscapes dataset [35]. It contains images of street scenes taken from a car with corresponding semantic segmentation maps. A total of 19 different semantic classes are labelled. Based on this dataset we designed a task that allows full control of the ambiguities: we create ambiguities by artificial random flips of five classes to newly introduced classes. We flip 'sidewalk' to 'sidewalk 2' with a probability of $8/17$, 'person' to 'person 2' with a probability of $7/17$, 'car' to 'car 2' with $6/17$, 'vegetation' to 'vegetation 2' with $5/17$ and 'road' to 'road 2' with probability $4/17$. This choice yields distinct probabilities for the ensuing $2^5 = 32$ discrete modes with probabilities ranging from 10.9% (all unflipped) down to 0.5% (all flipped). The official training dataset with fine-grained annotation labels comprises 2975 images and the validation dataset contains 500 images. We employ this offical validation set as a test set to report results on, and split off 274 images (corresponding to the 3 cities of Darmstadt, Mönchengladbach and Ulm) from the official training set as our internal validation set. As in the previous experiment, in this task we use a similar setting for the training processes of all approaches, which we present in detail in subsection H.2.

Fig. 3b shows samples of each approach in the comparison given one input image. In Appendix G we show further samples of other images, produced by our approach. Fig. 4b shows that the Probabilistic U-Net on the Cityscapes task outperforms the baseline methods when sampling 4, 8 and 16 times in terms of the energy distance. This edge in segmentation performance at 16 samples is highly significant according to the Wilcoxon signed-rank test ($p$-value $\sim \mathcal{O}(10^{-77})$). We have also conducted ablation experiments in order to explore which elements of our architecture contribute to its performance. These were (1) Fixing the prior, (2) Fixing the prior, and not using the context in the posterior and (3) Injecting the latent features at the beginning of the U-Net. Each of these variations resulted in a lower performance. Detailed results can be found in Appendix D.

**Reproducing the segmentation probabilities.** In the Cityscapes segmentation task, we can provide further analysis by leveraging our knowledge of the underlying conditional distribution that we have set by design. In particular we compare the frequency with which every model predicts each mode, to the corresponding ground truth probability of that mode. To compute the frequency of each mode by each model, we draw 16 samples from that model for all images in the test set. Then we count the number of those samples that have that mode as the closest (using 1-IoU as the distance function).

In Fig. 5 (and Figs. 8, 9, 10 in Appendix C) we report the mode-wise frequencies for all 32 modes in the Cityscape task and show that the Probabilistic U-Net is the only model in this comparison that is able to closely capture the frequencies of a large combinatorial space of hypotheses including very rare modes, thus supplying calibrated likelihoods of modes. The Image2Image VAE is the only

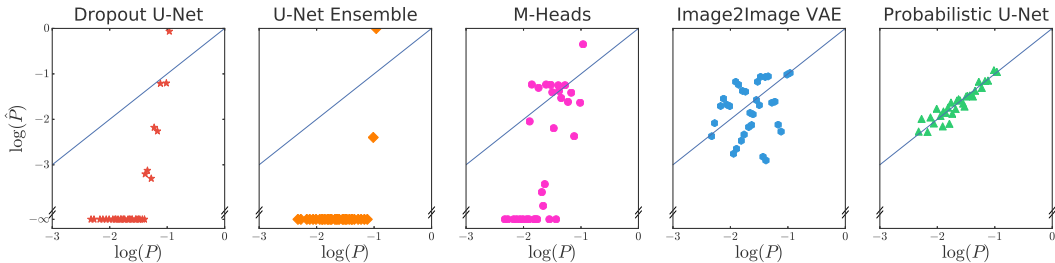

Figure 5: Reproduction of the probabilities of the segmentation modes on the Cityscapes task. The artificial flipping of 5 classes results in 32 modes with different ground truth probability (x-axis). The y-axis shows the frequency of how often the model predicted this variant in the whole test set. Agreement with the bisector line indicates calibration quality.

model among competitors that picks up on all variants, but the frequencies are far off as can be seen in its deviation from the bisector line in blue. The other baselines perform worse still in that all of them fail to represent modes and the modes they do capture do not match the expected frequencies.

### 4.3 Analysis of the Latent Space

The embedding of the segmentation variants in a low-dimensional latent space allows a qualitative analysis of the internal representation of our model. For a 2D or 3D latent space we can directly visualize where the segmentation variants get assigned. See Appendix A for details.

## 5 Discussion and conclusions

Our first set of experiments demonstrates that our proposed architecture provides consistent segmentation maps that closely match the multi-modal ground-truth distributions given by the expert graders in the lung abnormalities task and by the combinatorial ground-truth segmentation modes in the Cityscapes task. The employed IoU-based energy distance measures whether the models' individual samples are both coherent as well as whether they are produced with the expected frequencies. It not only penalizes predicted segmentation variants that are far away from the ground truth, but also penalizes missing variants. On this task the Probabilistic U-Net is able to significantly outperform the considered baselines, indicating its capability to model the joint likelihood of segmentation variants.

The second type of experiments demonstrates that our model scales to complex output distributions including the occurrence of very rare modes. With 32 discrete modes of largely differing occurrence likelihoods (0.5% to 10.9%), the Cityscapes task requires the ability to closely match complex data distributions. Here too our model performs best and picks the segmentation modes very close to the expected frequencies, all the way into the regime of very unlikely modes, thus defying mode-collapse and exhibiting excellent probability calibration. As an additional advantage our model scales to such large numbers of modes without requiring any prior assumptions on the number of modes or hypotheses.

The lower performance of the baseline models relative to our proposition can be attributed to design choices of these models. While the *Dropout U-Net* successfully models the pixel-wise data distribution (Fig. 8a bottom right, in the Appendix), such pixel-wise mixtures of variants can not be valid hypotheses in themselves (see Fig. 3). The *U-Net Ensemble*'s members are trained independently and each of them can only learn the most likely segmentation variant as attested to by Fig. 8b. In contrast to that the closely related *M-Heads* model can pick up on multiple discrete segmentation modes, due to the joint training procedure that enables diversity. The training does however not allow to correctly represent frequencies and requires knowledge of the number of present variants (see Fig. 9a, in the Appendix). Furthermore neither the U-Net Ensemble, nor the M-Heads can deal with the combinatorial explosion of segmentation variants when multiple aspects vary independently of each other. The *Image2Image VAE* shares similarities with our model, but as its prior is fixed and not conditioned on the input image, it can not learn to capture variant frequencies by allocating corresponding probability mass to the respective latent space regions. Fig. 17 in the Appendix shows a severe miss-calibration of variant likelihoods on the lung abnormalities task that is also reflected in its corresponding energy distance. Furthermore, in this architecture, the latent samples are fed into the U-Net's encoder path, while we feed in the samples just after the decoder path. This design choice in the Image2Image VAE requires the model to carry the latent information all the way through the U-Net core, while simultaneously performing the recognition required for segmentation, which might additionally complicate training (see analysis in Appendix D). Beside that, our design choice of late injection has the additional advantage that we can produce a large set of samples for a given image at a very low computational cost: for each new sample from the latent space only the network part after the injection needs to be re-executed to produce the corresponding segmentation map (this bears similarity to the approach taken in [23], where a generative model is employed to model hand pose estimation).

Aside from the ability to capture arbitrary modes with their corresponding probability conditioned on the input, our proposed *Probabilistic U-Net* allows to inspect its latent space. This is because as opposed to e.g. GAN-based approaches, VAE-like models explicitly parametrize distributions, a characteristic that grants direct access to the corresponding likelihood landscape. Appendix A discusses how the Probabilistic U-Net chooses to structure its latent spaces.

Compared to aforementioned concurrent work for image-to-image tasks [22], our model disentangles the prior and the segmentation net. This can be of particular relevance in medical imaging, where processing 3D scans is common. In this case it is desirable to condition on the entire scan, while retaining the possibility to process the scan tile by tile in order to be able to process large volumes with large models with a limited amount of GPU memory.

On a more general note, we would like to remark that current image-to-image translation tasks only allow subjective (and expensive) performance evaluations, as it is typically intractable to assess the entire solution space. For this reason surrogate metrics such as the inception score based on the evaluation via a separately trained deep net are employed [36]. The task of multi-modal semantic segmentation, which we consider here, allows for a direct and thus perhaps more meaningful manner of performance evaluation and could help guide the design of future generative architectures.

All in all we see a large field where our proposed Probabilistic U-Net can replace the currently applied deterministic U-Nets. Especially in the medical domain, with its often ambiguous images and highly critical decisions that depend on the correct interpretation of the image, our model's segmentation hypotheses and their likelihoods could 1) inform diagnosis/classification probabilities or 2) guide steps to resolve ambiguities. Our method could prove useful beyond explicitly multi-modal tasks, as the inspectability of the Probabilistic U-Net's latent space could yield insights for many segmentation tasks that are currently treated as a uni-modal problem.

## 6    Acknowledgements

The authors would like to thank Mustafa Suleyman, Trevor Back and the whole DeepMind team for their exceptional support, and Shakir Mohamed and Andrew Zisserman for very helpful comments and discussions. The authors acknowledge the National Cancer Institute and the Foundation for the National Institutes of Health, and their critical role in the creation of the free publicly available LIDC/IDRI Database used in this study.

## Footnotes

[3]An open source re-implementation of our approach can be found at `https://github.com/SimonKohl/probabilistic_unet`.

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
