[Supplementary Material]

# Appendix A Visualization of latent spaces

The segmentation variants from the proposed Probabilistic U-Net correspond to latent space samples from the learned prior distribution. Fig. 6 and Fig. 7 below show samples from the Probabilistic U-Net for an LIDC-IDRI and a Cityscapes example respectively. The samples are arranged so as to represent their corresponding position in a 2D-plane of the respective latent space. This allows to interpret how the model ends up structuring the space to solve the given tasks.

## A.1 Lung Abnormalities Segmentation

In the LIDC-IDRI case the $z_0$-component of the prior happens to roughly encode lesion size including a transition to complete lesion absence. The probability mass allocated to absence is relatively small in the particular example, which arguably is in tune with the fact that 1 of the 4 graders assessed the image as lesion free. The $z_1$-component on the other hand appears to encode shape variations. In the training, the posterior and the prior distribution are tied by means of the KL-divergence. As a consequence they 'live' in the same space and the graders (alongside the image to condition on) can be projected into the same latent space. Fig. 6 shows the grader's position in the form of green dots. The three graders that agree on presence, map into the 1-sigma interval of the prior, while the grader predicting absence falls just short of the 4-sigma isoprobability contour in the latent-space area that encodes absence. Fig. 3 gives more LIDC-IDRI examples with their corresponding grader masks and 16 random samples of the Probabilistic U-Net. It appears that our model agrees very well with cases for which there is inter-grader disagreement on lesion presence. For cases where the graders agree on presence, our model at times apparently shows an under-conservative prior, in the sense that uncertainty on presence can be elevated. The shape variations however are covered to a very good degree as attested by quantitative experiments above.

## A.2 Street Scene Segmentation

In the Cityscapes task we employ a latent space with more dimensions than on the lung abnormalities task in order to equip the prior with sufficient capacity to encode the grader modes. The best performing model used a 6D latent space, however, for ease of presentation the following discusses the latent structure of a 3D latent space version. Fig. 7 shows a $z_0$-$z_1$ plane of the latent space in which we again map corresponding segmentation samples, this time for a Cityscapes example. The precisely defined grader modes in the Cityscapes task can be identified with coherent regions in the latent space. As the space is 3D, not all 32 modes are fully manifest in the shown $z_2$-slice. The location of the modes is shown via white mode numbers and the degree of transparency indicated the proximity in $z_2$ relative to the shown slice. As this particular task involves discrete modes, the semantically different regions are coherent and well confined as hoped for. There however inevitably are transitions between those latent space regions that will translate to mixtures of the grader modes that cross over. Ideally these transitions are as sharp as possible relative to the order of magnitude of the prior variance, which is arguably the case. Fig. 18 shows Cityscapes examples with their corresponding grader masks and 16 random samples of the Probabilistic U-Net. The shown samples exhibit largely coherent variants alongside occasional variant mixtures that correspond to semantic cross overs in the latent space. As alluded to quantitatively before, the samples also appear to respect the grader variant frequencies, which are captured by structuring the latent-space under the prior in such fashion that the correct probability mass is allocated to the respective mode. In the upper boundary region of Fig. 7 improper samples are found that show miss-segmentations (although those are unlikely under the prior). The erroneously encoded modes found here are presumably attributable to the presence of inherent ambiguities in the dataset.

Figure 6: Visualization of the latent space for the lung abnormalities segmentation. $19 \times 19$ samples for a LIDC-IDRI test set example mapped to their prior latent-space position, using our model trained with a latent space of only 2 dimensions. For ease of presentation, the latent space is re-scaled so that the prior likelihood is a spherical unit-Gaussian. The isoprobable yellow circles denote deviations from the mean in sigma. The ground-truth grader masks' posterior position in this latent space is indicated by green numbers. The input image is shown in the lower left, to the right of it, the 4 grader masks are shown.

Figure 7: Visualization of the latent space for the Cityscapes task. $19 \times 19$ samples of a Cityscapes validation set example, mapped here to their latent-space position in the $z_0$-$z_1$ plane ($z_2 = 0$) of the learned prior, using our model trained with a latent space of only 3 dimensions. For ease of presentation, the samples are squeezed to rectangles and the latent space is re-scaled so that the prior likelihood is a spherical unit-Gaussian. The isoprobable yellow circles denote deviations from the mean in sigma. The ground-truth grader masks' posterior position in this latent space is indicated by white numbers. (color-map as in Fig. 18).

## Appendix B  Metrics

In the LIDC dataset, given that we have $m = 4$ ground truth samples and $n$ samples from the models, we employ the following statistic:

$$\hat{D}^2_{\text{GED}}(P_{\text{gt}}, P_{\text{out}}) = \frac{2}{nm} \sum_{i=1}^{n} \sum_{j=1}^{m} d(S_i, Y_j) - \frac{1}{n^2} \sum_{i=1}^{n} \sum_{j=1}^{n} d(S_i, S'_j) - \frac{1}{m^2} \sum_{i=1}^{m} \sum_{j=1}^{m} d(Y_i, Y'_j). \quad (6)$$

Here $d(x, y) = 1 - \text{IoU}(x, y)$, where $x$ and $y$ are the predicted and ground truth masks of the lesion. In the case that both are empty masks, we define its distance to be $0$, so that the metric rewards the agreement on lesion absence.

On the Cityscapes task, given that we have defined the settings, we have full knowledge about the ground truth distribution, which is a mixture of $M = 32$ Dirac delta distributions. Hence, we do not need to sample from it, but use it directly in the estimator:

$$\hat{D}^2_{\text{GED}}(P_{\text{gt}}, P_{\text{out}}) = \frac{2}{n} \sum_{i=1}^{n} \sum_{j=1}^{M} d(S_i, Y_j)\omega_j - \frac{1}{n^2} \sum_{i=1}^{n} \sum_{j=1}^{n} d(S_i, S'_j) - \sum_{i=1}^{M} \sum_{j=1}^{M} d(Y_i, Y'_j)\omega_i\omega_j, \quad (7)$$

where $\omega_j$ is the weight for the $j$-th mixture, which is a delta distribution containing all the density in $Y_j$. Here the distance $d$ depends on the average IoU of the 10 switchable classes only. Predicting one of such classes that is not present in the ground truth leads to a $0$ score, which will be one of the terms over which we average. The computed average does not account for classes that are not present in both prediction and ground truth.

## Appendix C  How models fit the ground truth distribution

In this section we analyse the frequency in which each mode of the Cityscape task is targeted by each model, and how much that varies from the ground truth distribution. We report the mode-wise and pixel-wise marginal occurrence frequencies of the sampled segmentation variants. In the mode-wise case, each sample is matched to its closest ground truth mode (using 1-IoU as the distance function). Then, the frequency of each mode is computed by counting the number of samples that most closely match that mode. In the pixel-wise case, the marginal frequencies $p(\text{predicted class}|\text{ground-truth class})$ are obtained by counting all pixels across all images and corresponding samples that show a valid pixel hypothesis given the ground-truth, normalized by the number of respective uni-modal ground-truth pixels. In Fig. 8 we present the results for U-Net Ensemble and Dropout U-Net, in Fig. 9 we show the results for M-Heads and Image2Image VAE, finally in Fig. 10 we present the results for our approach.

Figure 8: Reproduction of probabilities by the baselines Dropout U-Net and U-Net Ensemble. The vertical histogram shows the mode-wise occurrence frequencies of samples in comparison to the ground-truth probability of the modes, and the horizontal histogram reports the pixel-wise marginal frequencies, i.e. the sampled pixel-fractions for each new stochastic class (e.g. sidewalk 2) with respect to the corresponding existing one (sidewalk).

Figure 9: Reproduction of probabilities by the baselines M-Heads and Image2Image VAE. The vertical histogram shows the mode-wise occurrence frequencies of samples in comparison to the ground-truth probability of the modes, and the horizontal histogram reports the pixel-wise marginal frequencies, i.e. the sampled pixel-fractions for each new stochastic class (e.g. sidewalk 2) with respect to the corresponding existing one (sidewalk)

Figure 10: Reproduction of probabilities by our Probabilistic U-Net. The vertical histogram shows the mode-wise occurrence frequencies of samples in comparison to the ground-truth probability of the modes, and the horizontal histogram reports the pixel-wise marginal frequencies, i.e. the sampled pixel-fractions for each new stochastic class (e.g. sidewalk 2) with respect to the corresponding existing one (sidewalk).

# Appendix D    Ablation analysis

In this section we explore variations in the architecture of our approach, in order to understand how each design decision affects the performance. We have tried three variations over the original approach, these are:

**Fixing the prior**: Instead of making the prior a function of the context, here we fix it to be a standard Gaussian distribution.

**Fixing the prior, and not using the context (input image) in the posterior**: In addition to fixing the prior to be Gaussian, we also make the posterior a function of the ground truth mask only, ignoring the context.

**Injecting the latent features at the beginning of the U-Net**: Starting from our original model, we change the position in which the latent variables are used. Specifically here we concatenate them to the context (input image) and propagate that through the U-Net.

Figure 11: Ablation analysis. Comparison of architectural variations of our approach using the energy distance. Lower energy distances correspond to better agreement between predicted distributions and ground truth distribution of segmentations. The symbols that overlay the distributions of data points mark the mean performance.

In Fig. 11 we can observe that our approach is better than the other variations. As the mechanisms that induce the distributions over segmentations during sampling and training are blinded towards the context image, the performance in terms of the IoU-based energy distance decreases. In particular, our model is much better than the variation that injects latent samples at the beginning. This is a pleasant finding, given that our decision of injecting the latent variables at the end of the U-Net was motivated by efficiency reasons when sampling. Here we find that we do not lose performance by doing so, but instead observe an improved matching of the samples with the ground-truth distribution. We hypothesize that injecting the latent variables at the final stage of the pipeline makes it easier for the model to account for different segmentations given the same input. This hypothesis is supported by the slightly better performance shown by the alternative architecture when sampling only once, and how this advantage is lost, and actually reversed, when sampling several times.

# Appendix E    Predicting ground truth ambiguity from models' samples

In this section we assess the capacity of different models trained on LIDC for distinguishing between unambiguous and ambiguous instances. Specifically we define an instance to be ambiguous if 1 or more graders disagree on the presence of abnormal tissue. To do so, for each model we draw 16 samples per instance (as in all other experiments in the paper) and count the number of lesion presences out of the 16. This lesion presence is binned in two histograms with $[0, 16]$ bins, one for

ambiguous and one for unambiguous instances (they are shown in Fig. 12). Finally we evaluate the discriminatory power of such histograms by computing the best threshold that separates ambiguous and unambiguous instances on the validation set. We present the accuracy scores on the test set in Table 1, which shows the advantage that our approach has over the competitors in this regard.

Figure 12: Histograms showing the amount of (ground truth) ambiguous and unambiguous lesions as a function of the number of times the model produces a sample with a lesion in it (out of 16 samples). Each histogram corresponds to one model.

| Dropout U-Net | U-Net Ensemble | M-Heads | Image2Image VAE | Probabilistic U-Net |
|---|---|---|---|---|
| 0.328 | 0.699 | 0.678 | 0.678 | **0.736** |

Table 1: Discriminative power of histograms from different models to distinguish between ambiguous and unambiguous lesions.

# Appendix F   Sampling LIDC masks using different models

Fig. 13-17 show samples of our proposed model as well as all the baselines given the same input images. For reference the expert segmentations are shown in the four rows just below the images.

# Appendix G   Sampling Cityscapes segmentations using our model

Fig. 18 shows samples of our proposed model on the Cityscapes dataset, and Table 2 shows the numerical results from Fig. 4b, so that new approaches can be compared to those.

| # Samples | 1 | 4 | 8 | 16 |
|---|---|---|---|---|
| $\hat{D}^2_{\mathrm{GED}}$ | 0.874 | 0.337 | 0.248 | 0.206 |

Table 2: Numerical (mean) results of the Probabilistic U-Net on Cityscapes, taken from Fig. 4b.

Figure 13: Qualitative examples from the **Probabilistic U-Net**. The upper panel shows LIDC test set images from 15 different subjects alongside the respective ground-truth masks by the 4 graders. The panel below gives the corresponding 16 random samples from the network.

Figure 14: Qualitative examples from the **Dropout U-Net**. Same layout as Fig. 13.

Figure 15: Qualitative examples from the **U-Net Ensemble**. Same layout as Fig. 13.

Figure 16: Qualitative examples from the **M-Heads** (using a network with 16 heads). Same layout as Fig. 13.

Figure 17: Qualitative examples from the **Image2Image VAE**. Same layout as Fig. 13.

Figure 18: Qualitative examples from the **Probabilistic U-Net** on the Cityscapes task. The first row shows Cityscapes images, the following 4 rows show 4 out of the 32 ground truth modes with black pixels denoting pixels that are masked during evaluation. The remaining 16 rows show random samples of the network.

# Appendix H   Training details

In this section we describe the architecture settings and training procedure for both experiments.

## H.1   Lung abnormalities segmentation

We only use those lesions that were specified as a polygon (outline) in the XML files of the LIDC dataset, disregarding the ones that only have center of shape. That is, according to the LIDC paper we use the ones that are larger than 3mm, and filtering out the others, that are clinically less relevant [33]. We also filter out each Dicom file whose absolute value of SliceLocation differs from the absolute value of ImagePositionPatient[-1]. Finally we assume that two masks from different graders correspond to the same lesion if their tightest bounding boxes overlap.

During training image-grader pairs are drawn randomly. We apply augmentations to the image tiles ($180 \times 180$ pixels size): random elastic deformation, rotation, shearing, scaling and a randomly translated crop that results in a tile size of $128 \times 128$ pixels. The U-Net architecture we use is similar to [6] with the exception that we down- and up-sample feature maps by using bilinear interpolations. The cores of all models are identical and feature 4 down- and up-sampling operations, at each scale the blocks comprise three convolutional layers with $3 \times 3$-kernels, each followed by a ReLU-activation. In our model, both the prior and the posterior (as well as the posterior in Image2Image VAE) nets have the same architecture as the U-Net's encoder path, i.e. they are made up to the same number of blocks and type of operations. Their last feature maps are global average pooled and fed into a $1 \times 1$ convolution that predicts the Gaussian distributions parameterized by mean and standard deviation. The architecture last layers, corresponding to $f_{\mathrm{comb.}}$, comprise the appropriate number of $1 \times 1$-kernels and are activated with a softmax. The base number of channels is 32 and is doubled or respectively halved at each down- or up-sampling transition. All individual models share this core component and for ease of comparability we let all models undergo the same training schedule: the training proceeds over $240\,\mathrm{k}$ iterations with an initial learning rate of $1e^{-4}$ that is lowered to $1e^{-6}$ in 5 steps. All weights of all models are initialized with orthogonal initialization having the gain (multiplicative factor) set to 1, and the bias terms are initialized by sampling from a truncated normal with $\sigma = 0.001$. We use a batch-size of 32, weight-decay with weight $1e^{-5}$ and optimize using the Adam optimizer with default settings [37]. A KL weight of $\beta = 10$ with a latent space of 3 dimensions gave best validation results for the baseline Image2Image VAE, and $\beta = 1$ and a 6D latent space performed well for the Probabilistic U-Net, although the performances were alike across the hyperparameters tried on the validation set.

## H.2   Cityscapes

We down-sample the Cityscapes images and label maps to a size of $256 \times 512$. Similarly to above, we apply random elastic deformation, rotation, shearing, scaling, random translation and additionally impose random color augmentations on the images during training. The U-Net cores in this task are identical to the ones above, but process an additional feature scale (implying one additional up- and one additional down-sampling operation). The training procedure is also equivalent to the previous experiment, also using $240\,\mathrm{k}$ iterations, except that here we employ a batch-size of 16, and the initial learning rate of $1e^{-4}$ is lowered to $1e^{-5}$ in 3 steps. The Cityscapes dataset includes ignore label masks for each image with which we mask the loss during training, and the metric during evaluation. A KL weight of $\beta = 1$ and 3D latents gave best validation results for the Image2Image VAE and a $\beta = 1$ and 6D latents performed best for the Probabilistic U-Net (although 3-5D performed similarly).