[Reviews · NeurIPS 2018]

Reviewer 1



Post rebuttal: Authors have responded well to the issues raised, and I champion publication of this work. Main idea: Use a conditional variational auto-encoder to produce well-calibrated segmentation hypotheses for a given input. Strengths: The application is well motivated and experiments are convincing and state of the art. Convincing baselines, good discussion and conclusion. Nice illustrations in the appendix. Weaknesses: No significant theoretical contribution. Possibly in response, the manuscript is a little vague in its positioning relative to prior work. While relevant prior work is cited, the reader is left with some ambiguity and, if not familiar with this prior work, might be misled to think that there is methodological innovation beyond the specifics of architecture and application. Based on its solid and nontrivial experimental contribution I advocate acceptance; but the manuscript would profit from a clearer enunciation of the fact that / what prior work is being built on. Comments: 50: "treat pixels independently"; this is unclear (or faulty?), the quoted papers also use an encoder/decoder structure As a consequence, contribution 1 (line 74) is dubious or needs clarification. Contribution (2) is fine. Contribution (3): The statement is true, but the contribution is unclear. If the claim refers to the fact that latent z is concatenated only at the fully connected layers, then this has been done before (e.g. in DISCO Nets by Bouchacourt et al., (NIPS 2016)). Contribution (4): The claim is vague. If we take generative models in their generality, then it encompasses e.g. most of Bayesian Statistics and the claim of only qualitative evaluation is obviously wrong. If we only consider generative models in the deep learning world, the statement is correct insofar as many papers only contain qualitative evaluation of "My pictures are prettier than yours"; but there are nevertheless quantitative metrics, such as the inception score, or the metrics used in the Wasserstein AE. Section 2: It is not made very obvious to the reader which part of this VAE structure is novel and which parts are not. The paper does follow [4] and especially [5] closely (the latter should also be cited in line 82). So the only really novel part here is the U-net structure of P(y|z,x). Concatenating z after the U-net in (2) is new in this formulation, but not in general (e.g. as already mentioned by DISCO Nets (Bouchacourt at al., NIPS 2016)). Finally, there is no justification for the appearance of $\beta$ in (4), but it is up to the parameter name identical to what Higgins et al., (ICLR 2017) do with their \beta-VAE. Especially since authors choose $\beta >= 1$, which follows the Higgins et al. disentangling argument, and not the usual $\beta_t <= 1$, in which case it would be a time dependent downscaling of the KL term to avoid too much regularization in the beginning of the training (but then again the references to earlier work would be missing).

Reviewer 2



This paper focuses on the problem of image segmentation, addressing the specific issue of segmenting ambigious images for which multiple interpretations may be consistent with the image evidence. This type of problem may arise in medical domains, in which case awareness of this ambiguity would allow for subsequent testing/refinement, as opposed to simply predicting a single best segmentation hypothesis. With this motivation, this paper proposes a method for producing multiple segmentation hypotheses for a given potentially ambiguous image, where each hypothesis is a globally consistent segmentation. The approach taken is a combination of a conditional variational auto-encoder (CVAE) and U-Net CNN. Specifically: a prior net is used to model a latent distribution conditioned on an input image. Samples from this distribution are concatenated with the final activation layers of a U-Net, and used to produce a segmentation map for each sample. During training, a posterior network is used to produce a latent distribution conditioned on the input image and a given ground-truth segmentation. The full system is then trained by minimizing the cross-entropy between the predicted and ground-truth segmentations and the KL divergence between the prior and posterior latent distributions. The proposed method is evaluated on two different datasets - a lung abnormalities dataset in which each image has 4 associated ground-truth segmentations from different radiologists, and a synthetic test on the Cityscapes dataset where new classes are added through random flips (eg sidewalk class becomes "sidewalk 2" class with probability 8/17). Comparison is done against existing baseline methods for producing multiple segmentations, such as U-Net Ensemble and M-Heads (branching off last layer of network). Experiments show consistent improved performance using the proposed method, by evaluating the the predicted and ground-truth segmentation distributions using generalized energy distance. Additional analysis shows that the proposed method is able to recover lower probability modes of the underlying ground-truth distribution with the correct frequency, unlike the baseline methods. Overall, I found this to be a well-written paper with a nice method for addressing an important problem. Experimental validation is detailed and convincing. One small possible suggestion: part of the stated motivation for the paper is to allow for some indication of ambiguity to guide subsequent analysis/testing. It could be nice as an additional performance metric to have some rough evaluation in this practical context, as a gauge for an application-specific improvement. For instance, a criterion could be that lung images should be flagged for additional study if 1 or more of the experts disagree on whether the lesion is abnormal tissue; how often would this be correctly produced using the multiple predicted segmentations?

Reviewer 3



This paper deals with the problem of learning a distribution over segmentations given an input image. For that authors propose a generative segmentation model based on a combination of a U-Net with a conditional variational autoencoder that is capable of efficiently producing an unlimited number of plausible segmentation hypotheses. The problem is challenging, well-motivated and well presented. The related work is properly presented and the novelty of the proposed method is clear. The proposed method is original and means a sufficient contribution. The validation of the proposal and comparison with baseline methods is correct and the experimental results are quite convincing. In my opinion, the paper can be accepted as it is.